# Leptin Gene G2548A Polymorphism among Mongolians with Metabolic Syndrome

**DOI:** 10.3390/medsci7010003

**Published:** 2018-12-21

**Authors:** Batnaran Dagdan, Ariunbold Chuluun-Erdene, Orgil Sengeragchaa, Munkhzol Malchinkhuu, Munkhtsetseg Janlav

**Affiliations:** 1Department of Biochemistry and Laboratory Medicine, School of Biomedicine, Mongolian National University of Medical Sciences, Ulaanbaatar 14210, Mongolia; batnarandagdan@gmail.com (B.D.); ariunbold@mnums.edu.mn (A.C.-E.); Sengeragchaa.orgil@gmail.com (O.S.); 2Coronary Care Unit, Cardiovascular Center, The Shastin Central Hospital, Ulaanbaatar 16081, Mongolia; 3Department of Pathology, School of Biomedicine, Mongolian National University of Medical Sciences, Ulaanbaatar 14210, Mongolia.; munkhzol@mnums.edu.mn

**Keywords:** metabolic syndrome (MetS), leptin, *Lep*, *LepR*, single nucleotide polymorphism (SNP)

## Abstract

Metabolic syndrome (MetS) corresponds with multiple risk factors. Many studies have indicated that MetS significantly increases the risk of cardiovascular diseases and type 2 diabetes (T2D). The prevalence of MetS was estimated to be one third of the general Mongolian population in 2015. The purpose of our study was to determine polymorphisms of the *LEP* (Leptin) and *LEPR* (Leptin receptor) genes that show susceptibility to MetS and to predict the genetic risk of MetS. We selected 160 cases with MetS and 144 with healthy controls. The G2548A polymorphism of the *LEP* gene and the A668G (Q223R) polymorphism of the *LEPR* gene were genotyped using polymerase chain reaction-restriction fragment length polymorphism (PCR-RFLP). The results of the regression analysis showed that the 2548 amino acids (AA) of *LEP* gene carriers had increased incidences of MetS (OR = 3.23; *p* = 0.035). Patients with MetS who were 2548A allele carriers had an increased concentration of serum leptin (*p* = 0.011). Moreover, G2548A of *LEP* polymorphism was associated with elevated body mass index (BMI) and fasting blood glucose (FBG) in the case group. Our results confirm that the *LEP* G2548A loci is the independent risk factor of MetS.

## 1. Introduction

Metabolic syndrome (MetS) is defined by five components including insulin resistance, dyslipidemia (specifically, high triglycerides (TG) and low high-density lipoprotein (HDL)), central obesity, and hypertension [1,2]. A large number of longitudinal studies indicate that patients with MetS have significantly increased risks of developing cardiovascular diseases (CVD) and type 2 diabetes (T2DM) [3]. In 2006, the International Diabetes Federation (IDF) issued a scientific statement providing standardized clinical diagnostic guidelines and criteria for MetS [4]. Evaluated according to the IDF criteria, 32.7% of the general Mongolian population has MetS [5].

Genetic factors are believed to play an important role in the development of MetS [6,7]. Leptin is a protein product of the obesity (ob) gene or leptin (*LEP*) gene and is expressed and released by the adipose tissue in amounts proportional to body weight [8,9,10]. Studies on the leptin receptor (*LEPR*) have advanced the comprehension of the mechanism for regulating body weight and energy homeostasis and have shown that *LEPR* may influence the onset of obesity, T2DM, and other complex defects such as MetS [11,12,13].

Several single-nucleotide polymorphisms (SNPs) found in the *LEP* gene may be associated with serum leptin concentration or body mass index (BMI) [14,15,16,17]. Additionally, the *LEPR* genes have been investigated for gene variants potentially related to the pathophysiology of obesity, T2DM, and its associated complications [18]. Both the *LEP* and *LEPR* genes have been studied for polymorphisms that could possibly be related to the pathophysiology of obesity and its complications among specific ethnic groups [19,20,21]. However, the association of these polymorphisms with obesity is still controversial [21]. In addition, there have been contradictory findings regarding an association of MetS with the G2548A polymorphism of the *LEP* gene and with the A223G polymorphism of the *LEPR* gene [22,23,24].

The aim of this study was to seek predictable factors, such as frequencies of *LEP* and *LEPR* gene polymorphisms, to evaluate the risk of MetS among Mongolian subjects. Specifically, we intended to investigate whether the G2548A polymorphism of the *LEP* gene and the A223G polymorphism of the *LEPR* gene were involved with MetS among Mongolian subjects.

## 2. Materials and Methods

### 2.1. Study Subjects

We selected 160 patients with MetS (86 males and 74 females), aged 18–60 years old, from Ulaanbaatar, Mongolia for the case group. Metabolic syndrome was diagnosed based on a modified or harmonizing criteria proposed in 2009 by the International Diabetes Federation and the American Heart Association/National Heart, Lung, and Blood Institute [25], which was defined as the presence of any three of five risk factors from the following criteria: abdominal obesity with population- and country-specific cut points of waist circumference (WC) ≥90 cm for men and ≥80 cm for women, systolic blood pressure (SBP) ≥130 mmHg, diastolic blood pressure (DBP) ≥85 mmHg, serum TG level ≥150 mg/dl, serum HDL <40 mg/dl for men and serum HDL <50 for women, or fasting blood glucose (FBG) ≥100 mg/dl. Those with coronary heart disease, diabetes mellitus, or other chronic diseases, such as hepatic pathology, renal failure, dysthyroidism, hypertension and hyperlipidemia were excluded from the study. The control group consisted of 144 individuals (71 males and 73 females), aged 18–60 years old, with no history of obesity, hyperlipidemia, hypertension, or diabetes mellitus, verified by a health examination. Subjects receiving treatment for hypertension, hyperlipidemia, or hyperglycemia or taking any medicine that affects serum hormone measurement were thus exluded from the control group.

All participants voluntarily gave their informed consent, and the study was approved by the Ethics Committee of Mongolian National University of Medical Science (protocol #13–12/1A) and by the Ministry of Health Mongolia (protocol #7).

### 2.2. Biochemical Parameters

Biochemical parameters were analyzed for total cholesterol (TC), TG, HDL, and FBG using commercially available kits (AGAPPE Diagnostics Switzerland GmbH, Knonauerstrasse 54-6330, Cham, Switzerland). Low density lipoprotein (LDL) and homeostatic model assessment-insulin resistance (HOMA-IR) levels were measured using standard calculations [26,27]. Serum concentrations of adiponectin, leptin, and insulin were measured using a commercial direct enzyme-linked immunosorbent assay (ELISA) human adiponectin kit, a leptin kit, and an insulin kit, respectively (Linco Research, Inc., St. Louis, MO, USA).

### 2.3. Genotyping of Single-Nucleotide Polymorphisms

Two SNPs of the *LEP* and *LEPR* genes that were described in published studies indexed with the PubMed online searching system and the SNP database (the dbSNP National Center for Biotechnology Information) were selected as our study targets. Venous blood was collected from all participants after overnight fasting, and genomic DNA was extracted using the “G-spin Total DNA Extraction Kit” (iNtRON Biotechnology, Inc, Seongnam 13202, South Korea).

Single-nucleotide polymorphisms were genotyped by polymerase chain reaction and restriction fragment length polymorphism (PCR-RFLP) using the “Maxine PCR PreMix Kit” (i-Star Taq; iNtRON Biotechnology, Inc, Seongnam 13202, South Korea). For analysis of the *LEP* G2548A polymorphism, the following primers were used: forward primer—5′-TTTCTGTAATTTTCCCGTGAG-3′ and reverse primer—5′-AAAGCAAAGACAGGCATAAAAA-3′, and the PCR was carried out according to the previously published protocol [22]. For analysis of the *LEPR* Gln223Arg polymorphism, the following primers were used: forward primer—5′-ACCCTTTAAGCTGGGTGTCCCAAATAG-3′ and reverse primer—5′-AGCTAGCAAATATTTTTGTAAGCAATT-3′, and the reaction was conducted according to previously published protocol [28].

### 2.4. Statistical Analysis

Data analysis of our case-control study was conducted using SPSS 21.0 (IBM corporation, Chicago, IL, USA). The quantitative data was represented as the mean ± standard deviation (SD) and as the median (interquartile range). Statistical significance was evaluated using the *t*-test to compare the differences between two groups, while the Mann–Whitney U test was used for variables that were not normally distributed. The qualitative data were expressed in percentages, which was further analyzed using the χ^2^test. Genotype distributions of each SNP were compared between case and control groups using the χ^2^ test (3 × 2).

Allele distribution was determined using direct gene counting analysis and χ^2^ test (2 × 2). Multiple logistic regression analysis was performed to assess the effect of the SNP genotype on the development of MetS (with a 95% confidence interval).

For the association of polymorphism with lipid parameters, the genotype of significant polymorphisms was converted to a genetic model that constituted two groups: dominant homozygotes or non-risk allele carrier versus heterozygotes as Model 1 and recessive homozygotes or risk allele carriers as Model 2.

## 3. Results

The mean age of the patients with MetS was 41.7 ± 11.3 years and that of the controls was 41.2 ± 10.2 years. Of those with BMI ≥25 kg/m^2^, 64.1% were MetS patients and 35.9% were controls. Of those with WC ≥80 cm, 60.3% were MetS patients and 39.7% were controls. Of those with a serum TG level of ≥150 mg/dL, 88.9% were MetS patients and 11.1% were controls. No gender differences were found between the case and control groups. BMI, WC, SBP, and DBP levels were higher among those with MetS than among the controls.

### 3.1. Clinical Data and Biochemical Parameters

The clinical parameters of the MetS patients and the controls are summarized in Table 1. There was no difference in LDL levels between the MetS patients and the control group. The serum levels of TC, TG, and FBG were higher in the MetS group than in the control group (*p* = 0.028, *p* < 0.001), but HDL was lower in the MetS group than in the control group (*p* = 0.048). Insulin, HOMA-IR, and leptin levels were also higher in the MetS group (*p* < 0.05).

### 3.2. Allele Frequency of Single-Nucleotide Polymorphisms

The genotype frequencies of two SNPs in the MetS and control groups were calculated. The χ^2^ test revealed that the G2548A polymorphism of *LEP* gene, in Hardy–Weinberg equilibrium with both the MetS and control groups, was significantly associated with the prevalence of MetS (Table 2).

However, a significant difference was not observed in the allele frequency analysis. The multiple logistic regression model was used to determine the significance of the genotype of the SNP as a probable independent risk factor for MetS and found that the 2548AA homozygote genotype of *LEP* gene carriers had an increased risk for the development of MetS compared with those with the most frequent GG genotype (OR = 3.23; *p* = 0.035) (Table 3). After adjustments for age, gender, BMI, and WC, the increased risk for MetS was 2.78 times higher than in GG carriers.

With the *LEPR* A668G polymorphism, we found no statistically significant differences in the genotype frequencies, but the allele distribution differed between the MetS and control groups (*p* = 0.018).

In this study, we performed genotype combination analysis to evaluate the risk to MetS (Table 4). Our data analysis demonstrated that the genotype combination appears to confirm the SNP analysis with respect to G2548A and A668G polymorphisms in MetS. A total of nine available pairs were determined by the combination of the G223A and G2548A genotypes.

Of the nine possible pairs compared to the controls, a significant combined genotype related to the increased risk of MetS was not found in the combination analysis.

### 3.3. Association of Single-Nucleotide Polymorphisms with Biochemical Parameters

We evaluated the A allele of the G2548A SNP in the *LEP* gene using two models (Table 5). The analysis of Model 1 supported that, among those with MetS, leptin levels were higher in GA + AA carriers than in GG or non-carriers (*p* = 0.011). The same result was observed in Model 2, which analyzed the GG + GA and AA homozygote.

Additionally, GA + AA carriers had higher FBG levels than non-carriers in the Model 1 analysis (*p* = 0.014). However, this result was not found in Model 2. Moreover, Model 2 displayed that carriers of the AA homozygote of G2548A on the *LEP* gene promoter were more severely obese (BMI ≥ 36.5 ± 2.49) (*p* < 0.001).

## 4. Discussion

Over the past two decades, there has been a rapid increase in the number of people with MetS. This has been linked with devastating effects on human health, principally due to a higher risk for CVD, which is a leading cause of death [29]. Surveys of gene polymorphism carried out among various ethnic groups have suggested that genetic variants associated with the individual components of the MetS phenotype could contribute to its pathogenic mechanism [30]. Although the role of *LEP* and *LEPR* gene polymorphisms and their involvement in various diseases have been investigated extensively throughout the world in different populations, the association of *LEP* and *LEPR* variants with obesity, BMI, and MetS is still controversial [19,21,22,31]. To our knowledge, our study is the first attempt to test the association between *LEP* and *LEPR* polymorphisms and MetS in Mongolian subjects.

In this study, we analyzed the frequencies of the *LEP* G2548A and *LEPR* A668G polymorphisms among MetS patients in Mongolia. The allele frequency of the G2548A and A668G polymorphisms observed in our study was similar to that of other studies [19,22,32]. However, notable exceptions have been found in several studies on obesity patients among Taiwanese aborigines [33], Turkish patients [34], and MetS patients in Iran [23]. In these studies, the frequencies of the 2548G allele of the *LEP* gene were less than the A allele in the case groups. Furthermore, our study reported a strong association between the *LEP* G2548A genotype and MetS in the genotype comparison. This association was previously reported by Boumaiza et al. in Tunisia [22], but the *LEP* G2548A genotype was not observed as a risk factor for MetS in studies from Taiwan and Brazil [31,32]. These varying results for the G2548A genotypes in these studies may be due to ethnic and genetic differences or due to differences in the environmental conditions of the populations studied.

Hoffstedt et al. have suggested that the *LEP* G2548A variant may influence gene expression of leptin and leptin secretion by the adipose tissue [15]. The association of the *LEP* G2548A polymorphism and higher BMI levels, which was found in our study, was also reported in Tunisians with obesity [22]. However, we found that leptin levels were higher in GA + AA carriers than in GG or non-carriers among patients with MetS. A previous study also found increased BMI and leptin levels among G allele carriers [35]. Mammes et al. noted that the *LEP* G2548A polymorphism may influence an increase in BMI because of its effects on leptin secretion [14]. Hyperleptinemia is common in obese individuals who show leptin resistance, which is probably caused by impairing leptin signaling, a classical mechanism of hormone resistance [36]. This concept has been supported by rodent models [37]. Hence, obesity promotes hyperleptinemia, which, in turn, self-promotes leptin resistance and further promotes obesity, making leptin resistance both a consequence and cause of obesity [38]. Leptin is an adipocyte-derived signaling factor which has an important role in metabolic control, such as in stimulating glucose uptake, stimulating fatty acid oxidation, and reducing food intake [39].

Increasing leptin levels in Mongolian subjects can probably be explained by dietary intake patterns of consuming large quantities of protein, animal fat, and fatty dairy products, including milk [40]. In the last several decades, Mongolian society has been transformed from being based on nomadic animal husbandry to being based on settled industries in urbanized environments. Traditional Mongolian dietary intake has relied on meat, meat derivatives, and dairy products. However, this has now been replaced or supplemented with processed and extremely carbohydrate-rich foods. These transitions have significantly increased the risk of obesity among adults and consequently has resulted in an increasing prevalence of MetS in Mongolia [41,42]. With regards to obesity markers, BMI and WC values among Mongolian subjects in our study were higher than in other Asian populations, including Japanese and Koreans [43,44,45]. In contrast to our results, the *LEP* G2548A polymorphism demonstrated no link to leptin levels in Romanian and Egyptian subjects [21,46]. Additionally, no relationship between *LEP* G2548A and leptin levels were found in Melanesian and Micronesian Solomon Islanders [47]. Although the pathogenesis of MetS and each of its components is complex and although the molecular mechanism is still unclear, our findings suggest that the *LEP* G2548A polymorphism is associated with increased leptin levels, may represent a leptin resistance, and may take part in the development of MetS.

With regard to the A668G (Gln223Arg or Q223R) polymorphism of the *LEPR* gene, which results from the substitution of hydrophilic glutamine to hydrophobic arginine, we did not find an association with MetS, except for allele frequency. Due to that, we did not analyze the polymorphism with biochemical parameters. Interestingly, the A to G transition in exon 6 at nt 668 from the start codon 223 *LEPR* Q223R was associated with impaired leptin-binding activity [48]. Previous studies have reported that *LEPR* A668G is associated with impaired glucose tolerance or insulin resistance and T2DM [49,50]. It has also been related to increased leptin levels and been seen as a possible risk for MetS susceptibility [31,51].

Our findings indicate that the *LEP* G2548A polymorphism is a relevant MetS marker. Although the *LEP/LEPR* genotype combination was not observed to increase the risk of MetS when compared between the case and control groups for genetic interactions, the single genotype analysis indicated that carrying the *LEP* 2548AA is most likely to be involved with elevated BMI or obesity and with higher leptin concentrations. There are some limitations in this study, including small sample size, ethnicity, various environmental factors, changes in dietary patterns, and other genetic factors. In spite of the cross-sectional association of these SNPs, future studies are needed to investigate the association between a high number of SNPs per gene and genetic susceptibility to metabolic syndrome, including in massive cohorts from a Mongolian population. In conclusion, our study confirms that the G2548A polymorphism is the risk factor for the development of MetS because it is related to increasing leptin and BMI levels.

## Figures and Tables

**Table 1 medsci-07-00003-t001:** Main characteristics of the metabolic syndrome (MetS) and control groups.

Parameters	MetS (*n* = 160)	Control (*n* = 144)	*p*-Value
Age (years old)	41.7 ± 11.3	41.2 ± 10.2	0.924
Gender (M/F)	86/74	71/73	0.527
BMI (kg/m^2^)	31.27 ± 4.23	26.64 ± 3.75	<0.001
WC (cm)	100.97 ± 1.10	89.01 ± 12.75	<0.001
SBP (mmHg)	128.75 ± 13.97	114.02 ± 14.44	<0.001
DBP (mmHg)	88.42 ± 9.92	77.95 ± 9.52	<0.001
FBG (mg/dL)	92.07 ± 66.57	71.69 ± 12.69	0.012
TG (mg/dL) ^1^	124.45 (83.74–179.37)	66.37 (49.52–96.35)	<0.001
TC (mg/dL)	157.72 ± 36.42	148.47 ± 36.73	0.123
HDL-C (mg/dL)	32.04 ± 11.49	36.44 ± 15.63	0.048
LDL-C (mg/Dl)	96.85 ± 40.38	95.75 ± 39.53	0.867
Insulin (ng/mL) ^1^	12.06 (0.29–113.53)	8.53 (0.29–170.29)	0.015
HOMA-IR ^1^	2.28 (0.05–38.99)	1.43 (0.05–36.32)	0.006
Adiponectin (ng/mL) ^1^	6.46 (0.06–19.51)	6.18 (0.09–49.66)	0.936
Leptin (ng/mL) ^1^	11.10 (2.30–56.30)	4.5 (0.01–34.87)	<0.001

BMI, body mass index; M, male, F, female; WC, waist circumference; SBP, systolic blood pressure; DBP, diastolic blood pressure; FBG, fasting blood glucose; TG, triglyceride; TC, total cholesterol; HDL, high-density lipoprotein; LDL, low-density lipoprotein; HOMA-IR, homeostatic model assessment-insulin resistance. Values for continuous variables are expressed as the mean ± standard deviation (SD) and as the median and interquartile range (IQR). ^1^ The Mann–Whitney U test was used for those variables. The *t*-test was utilized for normally distributed variables.

**Table 2 medsci-07-00003-t002:** Genotype frequencies of single nucleotide polymorphisms (SNPs).

Gene	SNPs	Genotype	MetS Group % (*n*)	Control Group % (*n*)	*p*-Value
*LEP*	G2548A	GG	61.2% (98)	68.7% (99)	0.035
		GA	33.12% (53)	29.8% (43)	
		AA	5.6% (9)	1.4% (2)	
		G	77.8% (249)	83.7% (241)	0.104
		A	21.2% (71)	16.3% (47)	
*LEPR*	A668G (Gln223Arg)	AA	69.4% (111)	56.9% (82)	0.053
		AG	26.8% (43)	36.1% (52)	
		GG	3.7% (6)	6.9% (10)	
		A	82.81% (265)	75.0% (216)	0.018
		G	17.19% (55)	25.0% (72)	

Genotype distributions of each SNP were compared between case and control groups using the χ_2_ test (3 × 2) and (2 × 2).

**Table 3 medsci-07-00003-t003:** Genotype frequencies of polymorphisms associated with MetS.

Gene/SNP	Genotype	MetS Group % (*n*)	Control Group % (*n*)	OR (95% CI) ^1^	OR (95% CI) ^2^
*L**EP* G2548A	GG	61.2% (98)	68.7% (99)	1	1
	GA	33.12% (53)	29.8% (43)	1.22 (0.69–1.86)	1.31 (0.62–1.61)
	AA	5.6% (9)	1.4% (2)	3.23 * (1.30–18.51)	2.78 * (1.06–7.59)
	GG/GA + AA	38.7% (62)	31.25% (45)	2.18 (0.94–3.16)	1.72 (0.63–2.13)
	GG+GA/AA	7.50% (9)	1.39% (2)	2.94 (1.08–17.25)	2.16 (0.93–7.16)

SNP, single nucleotide polymorphism; OR, odds ratio; CI: Confidence Interval. ^1^ Adjusted with BMI and WC; ^2^ adjusted with age, gender, BMI and WC. * *p* < 0.05.

**Table 4 medsci-07-00003-t004:** Genotype combination of *LEP* and *LEPR* gene polymorphisms for MetS.

Combination	LepR G223A	Lep G2548A	Frequency % (*n*)		OR ^1^ (95% CI)	*p*-Value
			MetS (*n* = 160)	Control (*n* = 144)		
1	AA	GG	43.7 (70)	38.9 (55)	1.03 (0.42–1.78)	0.487
2	AA	GA	20 (31)	0.16 (24)	1.12 (0.38–1.82)	0.245
3	AA	AA	6.25 (10)	0.01 (2)	2.86 (1.0–20.79)	0.051
4	AG	GG	0.15 (24)	0.23 (35)	0.51 (0.29–1.47)	0.062
5	AG	GA	0.10 (16)	0.12 (18)	0.64 (0.31–1.50)	0.845
6	AG	AA	0.025 (3)	-	-	-
7	GG	GG	0.025 (4)	0.05 (8)	0.36 (0.09–1.32)	0.285
8	GG	GA	0.01 (1)	0.01 (2)	0.74 (0.05–1.84)	0.113
9	GG	AA	-	-	-	-

^1^ Multiple logistic regression analysis was performed with adjustment for age, gender, BMI, and WC.

**Table 5 medsci-07-00003-t005:** Comparison between the genotype of the LEP G2548A polymorphism and clinical features in the MetS group.

Clinical Features	Model 1			Model 2		
	GG	GA + AA	*p*-Value	GG + GA	AA	*p*-Value
BMI (kg/M^2^)	31.13 ± 4.00	31.49 ± 4.56	0.608	30.85 ± 4.04	36.52 ± 2.49	<0.001
WC (cm)	99.30 ± 10.37	102.03 ± 11.41	0.130	100.53 ± 11.24	106.41 ± 6.94	0.077
BP, systolic (mmHg)	129.69 ± 15.17	127.25 ± 11.65	0.283	128.64 ± 13.90	130.0 ± 14.77	0.748
BP, diastolic (mmHg)	87.65 ± 10.89	89.64 ± 7.98	0.216	88.17 ± 10.05	91.50 ± 7.22	0.264
TC (mg/dL)	156.70 ± 39.08	159.29 ± 31.78	0.663	157.76 ± 36.75	157.20 ± 36.56	0.960
TG (mg/dL) ^1^	129.20 (89.61–183.1)	105.50 (81.09–174.8)	0.828	128.95 (86.74–174.80)	101.76 (75.45–185.40)	0.697
HDL (mg/dL)	33.89 ± 13.26	30.86 ± 10.05	0.104	34.02 ± 13.82	33.27 ± 10.75	0.681
LDL (mg/dL)	96.90 ± 36.95	93.97 ± 43.18	0.650	96.12 ± 38.78	96.05 ± 32.98	0.957
FBG (ng/mL) ^1^	75.37 (60.50–80.57)	77.95 (68.13–95.46)	0.014	75.59 (65.91–88.40)	80.65 (70.78–107.90)	0.228
Adiponectin (ng/mL) ^1^	6.33 (0.060–19.51)	6.51 (1.34–12.91)	0.572	6.30 (4.48–8.93)	8.26 (6.36–10.61)	0.067
Leptin (ng/mL) ^1^	9.5 (2.3–38.0)	16.1 (3.5–56.3)	0.011	9.7 (6.7–21.6)	43.65 (36.0–45.8)	0.001
Insulin ^1^ (µIU/mL)	12.06 (0.29–84.71)	12.94 (1.18–113.53)	0.861	11.76 (7.35–1.71)	13.97 (13.24–95.0)	0.120
HOMA-IR ^1^	2.37 (0.05–38.99)	2.20 (0.20–25.31)	0.344	2.24 (1.19–4.0)	8.4 (1.64–22.77)	0.108

Values for continuous variables are expressed as the mean ± standard deviation (SD) and as median and interquartile range (IQR). ^1^ The Mann–Whitney U test was used for those variables. The *t*-test was utilized for normally distributed variables.

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
