# Peer review of "Leptin Gene G2548A Polymorphism among Mongolians with Metabolic Syndrome"

_medsci, 2018, doi:10.3390/medsci7010003_

Reviewer 1 Report

Overall comments

In the current study, Dagdan et al. investigated the association of the leptin (LEP) and leptin receptor (LEPR) gene polymorphism with the metabolic syndrome (MetS) among Mongolian subjects. The frequencies of the G2548A of the LEPand the A668G of the LEPR gene was characterized in 160 subjects with MetS and 144 healthy controls. The G2548A polymorphism was significantly associated with the prevalence of MetS in Chi2 test. Patients with MetS with 2548A allele had increased serum level of leptin. Multiple logistic regression model showed that carries of the 2548AA LEPgene had an increased incident of MetS (OR=3.23, p=0.035). Based on these findings, the authors concluded that the LEPG2548A loci is the independent risk factor of MetS. Overall, the manuscript is of interest and addresses an important health topic regarding the risk of MetS and its association with leptin gene polymorphism. 

Specific comments

1.    Abstract: third sentence stating that the prevalence of MetS is 32.7% in general Mongolian population should be rephrased. It could state in 2006 the prevalence of MetS among Mongolians was ~30% as defined by the IDF definition.  

2.    Study objectives are vaguely defined in the Abstract and in the Introduction parts. 

3.    In the 'Study subjects' section please provide more details on inclusion and exclusion criteria.

4.    In the definition of MetS, the authors state that MetS was diagnosed using the IDF criteria, which is “as the presence of at least three of the following criteria”. According to the IDF definition, for a person to be defined as having a metabolic syndrome, they must have: Central obesity (defined as WC with ethnicity specific values), plus any two of the following factors (high TG, low HDL cholesterol, raised blood pressure and raised fasting plasma glucose). So, it’s not clear whether the authors actually used the IDF criteria, it sounds like that the NCEP ATP-III definition of MetS was used. Please clarify whether the ethnicity specific values were used for central obesity and add proper citation.   

5.    The description of the statistical analysis needs clarification. Please clarify whether the distribution of all variables was tested for normality, and non-normally distributed variables were appropriately transformed prior analyses (insulin, triglycerides, HOMA-IR, adiponectin, leptin). Please provide details of multiple logistic regression analysis, dependent and independent variables included in the model and covariates. It seems that the model was “double” adjusted for BMI and WC (footnote on table 3 and 4), please explain why only obesity was adjusted in the model. It would be interesting to see the model adjusted for age, gender, smoking etc. 

6.    Genotype frequency for GA and AA of G2548A SNP in the MetS group is different in Table 2 and 3, please clarify.  

7.    Table 4. Title of the table is odd. Also the Combination 1-9 in table might be confusing for the readers, please provide a clear explanation in the text. 

8.    The limitations should include that this study only examined the cross-sectional association of LEP and LEPR gene polymorphism with metabolic syndrome and its components. 

Author Response

1.    Abstract: third sentence stating that the prevalence of MetS is 32.7% in general Mongolian population should be rephrased. It could state in 2006 the prevalence of MetS among Mongolians was ~30% as defined by the IDF definition.  

Response: we rephrased this sentense in introduction according your recommendation.

2.    Study objectives are vaguely defined in the Abstract and in the Introduction parts. 

Response: we revised objectives in abstract and introduction sections again. The edited version was written in track changes.

3.    In the 'Study subjects' section please provide more details on inclusion and exclusion criteria.

Response: We inclusion and exclusion criteria were revised again.

4.    In the definition of MetS, the authors state that MetS was diagnosed using the IDF criteria, which is “as the presence of at least three of the following criteria”. According to the IDF definition, for a person to be defined as having a metabolic syndrome, they must have: Central obesity (defined as WC with ethnicity specific values), plus any two of the following factors (high TG, low HDL cholesterol, raised blood pressure and raised fasting plasma glucose). So, it’s not clear whether the authors actually used the IDF criteria, it sounds like that the NCEP ATP-III definition of MetS was used. Please clarify whether the ethnicity specific values were used for central obesity and add proper citation.   

Response: We made a mistake for indicating the criteria of MetS definition into study subjects section in our manuscript. Actually, we used the harmonizing criteria proposed by the International Diabetes Federation and the American Heart Association/National Heart, Lung, and Blood Institute in 2009. According your recommendation, we revised those criteria again, explained measurement values as exact as harmonizing criteria criteria. The waist circumference should be measured by ethnicity or population specific values. Thus, we select a cut point of WC in Asia according to recommendation of WHO.

5.    The description of the statistical analysis needs clarification. Please clarify whether the distribution of all variables was tested for normality, and non-normally distributed variables were appropriately transformed prior analyses (insulin, triglycerides, HOMA-IR, adiponectin, leptin). Please provide details of multiple logistic regression analysis, dependent and independent variables included in the model and covariates. It seems that the model was “double” adjusted for BMI and WC (footnote on table 3 and 4), please explain why only obesity was adjusted in the model. It would be interesting to see the model adjusted for age, gender, smoking etc. 

Response: we prefered to use an appropriate statistic tests instead of transforming non normal distributed data (insulin, triglycerides, HOMA-IR, adiponectin, leptin) into distributed data by using arithmetical logarithm. Depending on the distribution of continuous data, we used t-test for normal distribution in order to compare the differences between two groups while Mann-Whitney U test was utilized for non normal distributed data. We made  interpretations of each statistic tests in tables however we have already noted it in “Statistical analysis” section.

As regards to multiple logistic regression analysis, dependent variable was MetS (yes and no), independent variables were Lep G2548A (GG, GA and AA) or recessive (GG+GA/AA) and dominant (GG/GA+AA) model. Why we have adjusted without age and gender was that there was no significant difference for age and gender between case and control group. Therefore, in regression model analysis, age and gender were not calculated as more effective factors as obesity (BMI and WC) values in our study population. But, we followed your recommendations and recalculated the logistic regression analysis in adjustment with age, gender, BMI and WC. About smoking status, we recorded data without questionnaire of smoking. So, it would be an unable to estimate with smoking frequencies.

6.    Genotype frequency for GA and AA of G2548A SNP in the MetS group is different in Table 2 and 3, please clarify.  

Response: we made technical mistake, then corrected the frequencies of genotype.

7.    Table 4. Title of the table is odd. Also the Combination 1-9 in table might be confusing for the readers, please provide a clear explanation in the text. 

Response: we also made technical mistake on title of the table 4, then corrected the table. We added detailed explanation to results text related to table 4.  

8.    The limitations should include that this study only examined the cross-sectional association of LEP and LEPR gene polymorphism with metabolic syndrome and its components.

        Response: as following your advice, we inserted concept about cross-sectional association         of LEP and LEPR gene polymorphism  in the study limitation part.

Reviewer 2 Report

There is an interested but some improvement is needed.

Editorial work and language corrections are still needed, e.g.

 L.52: should be “contradictory relations or data” not “evidence”. Contradictory data indicate no evidence

L.54 – should be :the aim of the study” not “significance…”

L. 57 – should be “were” not “was”

Please go through the manuscript and make corrections.

Why 160subjects  for case group and 144 for control group were chosen? Did the Authors perform the power analysis? The power analysis is of great importance to confirm the suggestion that “the LEP G2548A loci is the independent risk factor of MetS”

It should be clearly indicated in the text which differences were found based on adjusted analyses.

Table 5 – It is clear that AA homozygotes had significantly higher BMI than G allele carriers, and significantly higher leptin levels is this group is probably the result of BMI or at least significantly affected by BMI. Therefore, this should be taken into account when leptin data of model 1 are discussed.

Author Response

L.52: should be “contradictory relations or data” not “evidence”. Contradictory data indicate no evidence

Response: we replaced the “contradictory evidence” with “contradictory relations” in line 52, according to your recommendation

L.54 – should be :the aim of the study” not “significance…”

Response: we changed the “significance” with “the aim of the study” in line 54, according to your recommendation

L. 57 – should be “were” not “was”

Response: we corrected the grammar error

Please go through the manuscript and make corrections.

Why 160subjects  for case group and 144 for control group were chosen? Did the Authors perform the power analysis? The power analysis is of great importance to confirm the suggestion that “the LEP G2548A loci is the independent risk factor of MetS”

Response: We agree with you about power of sample size in case-control study. We have estimated sample size of our study by using sample size formula and online sample size calculator (www.clinCalc.com) before we started. When the power of study was 0.9 or 90%, α was 0.05 or 5% and an influencing factor was at least 2 or OR=2, the sample size was calculated as 100 in minimum for each group. Therefore, we intended to reach higher amount of sample size than observed quantity (n=100) and we selected 160 patients for case group and 144 subjects for control group based on sample size calculation.

It should be clearly indicated in the text which differences were found based on adjusted analyses.

Response: we made two types adjustent in regression analysis and noted this result in the text

1.      Adjusted with BMI and WC

2.      Adjusted with age, gender, BMI and WC

Table 5 – It is clear that AA homozygotes had significantly higher BMI than G allele carriers, and significantly higher leptin levels is this group is probably the result of BMI or at least significantly affected by BMI. Therefore, this should be taken into account when leptin data of model 1 are discussed.

Response: we accepted your recommendation.